# Direct Detection of Glycated Human Serum Albumin and Hyperglycosylated IgG3 in Serum, by MALDI-ToF Mass Spectrometry, as a Predictor of COVID-19 Severity

**DOI:** 10.3390/diagnostics12102521

**Published:** 2022-10-17

**Authors:** Ray K. Iles, Jason K. Iles, Jonathan Lacey, Anna Gardiner, Raminta Zmuidinaite

**Affiliations:** 1MAP Sciences, The iLab, Stannard Way, Bedford MK44 3RZ, UK; 2Laboratory of Viral Zoonotics, Department of Veterinary Medicine, University of Cambridge, Madingley Road, Cambridge CB3 0ES, UK; 3NISAD, Sundstorget 2, 252-21 Helsingborg, Sweden

**Keywords:** COVID-19, MALDI-ToF MS, plasma, glycated albumin, glycovariant/glycated IgG3

## Abstract

The prefusion spike protein of SARS-CoV-2 binds advanced glycation end product (AGE)-glycated human serum albumin (HSA) and a higher mass (hyperglycosylated/glycated) immunoglobulin (Ig) G3, as determined by matrix assisted laser desorption mass spectrometry (MALDI-ToF). We set out to investigate if the total blood plasma of patients who had recovered from acute respiratory distress syndrome (ARDS) as a result of COVID-19, contained more glycated HSA and higher mass (glycosylated/glycated) IgG3 than those with only clinically mild or asymptomatic infections. A direct serum dilution, and disulphide bond reduction, method was developed and applied to plasma samples from SARS-CoV-2 seronegative (*n* = 30) and seropositive (*n* = 31) healthcare workers (HCWs) and 38 convalescent plasma samples from patients who had been admitted with acute respiratory distress (ARDS) associated with COVID-19. Patients recovering from COVID-19 ARDS had significantly higher mass AGE-glycated HSA and higher mass IgG3 levels. This would indicate that increased levels and/or ratios of hyper-glycosylation (probably terminal sialic acid) IgG3 and AGE glycated HSA may be predisposition markers for the development of COVID-19 ARDS as a result of SARS-CoV2 infection. Furthermore, rapid direct analysis of serum/plasma samples by MALDI-ToF for such humoral immune correlates of COVID-19 presents a feasible screening technology for the most at risk; regardless of age or known health conditions.

## 1. Introduction

As the COVID-19 pandemic has gripped the world, the medical technology strategy to deal with the dynamic situation has evolved to include mass screening and mass vaccination. In addition, extensive investigations to identify individuals at risk of severe disease has been undertaken.

Mass screening has proved challenging given the logistics involved and the need for affordable, easy repeat testing and track and trace [1]. Vaccination has made a huge impact on hospitalisations rates in countries where over 70% of the adult population have been immunised [2]. Booster vaccinations are likely needed to deal with emerging variant forms of SARS-CoV-2 [3]. A huge amount of money, time and national resources have been invested in medical technologies to meet the ongoing demand for diagnostics. Given that the infection with SARS-CoV-2 virus was life threatening to a minority of the population [4,5] and that vaccination has reduced this risk further; social interaction, business, trade and international travel have reopened [6].

However, clinicians and scientist are seeking to identify those who are at the greatest risk of severe disease to help inform and rationalise targeted testing and booster vaccinations [7]. Aside from respiratory disorders and heart disease; age, obesity and diabetes appear to be the major clinical risk conditions associated with developing COVID-19 clinical syndromes and death as a result of SARS-CoV2 infection [8,9].

A range of medical technologies are being explored and developed to help inform a more targeted or precision medicine approach. Currently, the focus has been on artificial intelligence (AI) identifying the “at risk” from medical notes, and pre-existing test results, via algorithms [10,11,12]. However, it is questionable whether this will be sufficient to give the level of precision (sensitivity and specificity) required in identifying all that are at risk. The fault will lay not with the machine learning (ML) or AI; but that insufficient robust correlating test results are being provided to the algorithms. New more significant and non-subjective evaluations are required, such as bio-markers and immune correlates [13].

We have recently shown that the prefusion spike protein of SARS-CoV-2 binds glycated albumin and a higher molecular mass of IgG3; due to glycation or hyper glycosylation. We proposed this benefits the virus immune evasion strategies and may contribute to COVID-19 pathologies as a result of a selective cloud of serum proteins triggering thrombolytic and detrimental inflammatory pathways [14,15]. Furthermore, an increased ratio of IgG3 compared to IgG1 in the humoral response to SARS-CoV-2 is associated with an adverse outcome of a COVID-19 infection [16]. The possibility is that elevated glycated albumin, along with elevated levels of IgG3 characterised by an increased molecular mass (due to glycation/glycosylation), may act as potential markers of susceptibility to develop ARDS as a result of SARS-CoV-2 infection.

We examined the mass spectra of samples from convalescent patients and sero-negative HCWs, without immunocapture, to see if a direct MALDI-ToF analysis of diluted and disulphide reduced blood serum/plasma would reveal gross characteristic differences in resolved immunoglobulin heavy chains (Hc) and (glycated) albumin. Given the logistics of simple pin prick blood home sampling, high throughput, low cost, MALDI-ToF mass spectrometry could provide one of the marker tests for the COVID-19 “at risk” individuals.

## 2. Materials and Methods

### 2.1. Samples

Serum and plasma samples were obtained from HCWs and COVID-19 patients referred to the Royal Papworth Hospital for critical care. COVID-19 patients hospitalised during the first wave and as well as NHS healthcare workers working at the Royal Papworth Hospital in Cambridge, UK served as the exposed HCWs cohort (Study approved by Research Ethics Committee Wales, IRAS: 96194 12/WA/0148. Amendment 5). NHS HCWs participants from the Royal Papworth Hospital were recruited through staff email over the course of 2 months (20 April 2020–10 June 2020) as part of a prospective study to establish seroprevalence and immune correlates of protective immunity to SARS-CoV-2. Patients were recruited in convalescence either pre-discharge or at the first post-discharge clinical review. All participants provided written, informed consent prior to enrolment in the study. Sera from NHS HCWs and patients were collected between July and September 2020, approximately three months after they were enrolled in the study.

For cross-sectional comparison, representative convalescent serum and plasma samples from seronegative HCWs, seropositive HCWs and convalescent polymerase chain reaction (PCR)-positive COVID-19 patients were obtained. The serological screening to classify convalescent HCWs as positive or negative was done according to the results provided by a CE-validated Luminex assay detecting Nucleocapsid (N), receptor binding domain (RBD) and Spike (S) specific IgG; a lateral flow diagnostic test (IgG/IgM) and an electro-chemiluminescence assay (ECLIA) detecting N- and S-specific IgG. Any sample that produced a positive result by any of these assays was classified as positive. Thus, the panel of convalescent plasma samples (3 months post-infection) were grouped in three categories: (A) Seronegative Staff (N = 30 samples) (B) Seropositive Staff (N = 31 samples); (C) Patients (N = 38 samples) [17].

### 2.2. Sample Analysis by MALDI-ToF Mass Spectrometry

Mass spectra were generated using a 15 mg/mL concentration of sinapinic acid (SA) matrix. One µL of plasma was diluted in 40 µL of ultra-pure mass spec grade water (Romil Ltd., Water beach Cambridgeshire, UK). After thorough vortex mixing, 10 µL of the diluted plasma was mixed with mixed 1:1 with 10 µL of 20 mM tris(2-carboxyethyl)phosphine (TCEP) (Sigma-Aldrich, Poole, UK) in ultra-pure water. After incubation for 15 min at room temperature 1 µL of the diluted samples were taken and plated on a 96 well stainless-steel target plate using a sandwich technique [18]. The MALDI-ToF mass spectrometer (microflex^®^ LT/SH, Bruker, Coventry, UK) was calibrated using a 2-point calibration of 2 mg/mL bovine serum albumin (2 + ion 33,200 *m*/*z* and 1 + ion 66,400 *m*/*z*) (Pierce™, ThermoFisher Scientific, Waltham, MA, USA). Mass spectral data were generated in a positive linear mode. The laser power was set at 65% and the spectra was generated at a mass range between 10,000 to 200,000 *m*/*z*; pulsed extraction set to 1400 ns.

A square raster pattern consisting of 15 shots and 500 positions per sample was used to give 7500 total profiles per sample. An average of these profiles was generated for each sample, giving a reliable and accurate representation of the sample across the well. The raw, averaged spectral data was then exported in a text file format to undergo further mathematical analysis.

### 2.3. Spectral Data Processing

Mass spectral data generated by the MALDI-ToF instrument were uploaded to an open-source mass spectrometry analysis software mMass™ [19], where it was processed by using; a single cycle, Gaussian smoothing method with a window size of 300 *m*/*z*, and baseline correction with applicable precision and relative offset depending on the baseline of each individual spectra. In the software, an automated peak-picking was applied to produce peak lists which were then tabulated and used in subsequent statistical analysis.

### 2.4. Statistical Analysis

Peak mass and peak intensities were tabulated in excel and plotted in graphic comparisons of distributions for each antigen-capture and patient sample group. Means and medians were calculated and, given the asymmetric distributions found, non-parametric statistics were applied, such as Mann–Whitney U test, when comparing differences in group distributions.

## 3. Results

The developed direct plasma sample MALDI-ToF mass spectra analysis method was rapid and required minimal reagents. Optimised pulse extraction allowed for the IgG subtypes to be resolved in most samples. From previous spectra calibration using pure preparation, IgG1 Hc averaged at peak apex ~51,000 *m*/*z*, IgG3 at ~54,000 *m*/*z*, IgA at ~56,000 *m*/*z* and IgM at ~74,000 *m*/*z*. Human albumin was at 66,400 *m*/*z* (1+) and transferrin at 79,600 *m*/*z*. All are broad heterogenous peaks reflecting glycosylation/glycation and sequence variation, and were identifiable in the plasma sample spectra (see Figure 1). A further as yet to be confirmed Ig heavy chain (IgX), presumed to be IgG4, was often seen resolving at ~48,000 *m*/*z* [14,15].

### 3.1. Direct Plasma Total IgG1 Spectra

IgG1 Hc was detected in all samples and was the most dominant and abundant immunoglobulin, such that less abundant IgG3 and the related Ig termed IgX Hc were often masked under the broad polyclonal IgG1 peak. Although intensity levels varied, no significant association with the COVID-19 sample sets was found. Similarly, variance in IgG1 average molecular mass, although slightly lower for the ARDS COVID-19 patient set, was not statistically significant: Mean difference Δ31 *m*/*z*, −0.06% change in mass, not significant *p* = 0.177 (See Figure 2).

### 3.2. Direct Plasma Total IgG3 Spectra

IgG3 Hc mass spectral peaks were detected in 43/99 samples and showed an increasing intensity in those who had been infected with SARS-CoV-2, with highest levels being found in the COVID-19 ARDS patients’ convalescent plasma. In the sero-negative HCWs samples, peaks were detected in 37% of samples, mean intensity 30 arbitrary units (AU); in the sero-positive HCWs samples, peaks were detected in 32% of samples, mean intensity 27AU and in the COVID-19 ARDS patient samples, peaks were detected in 58% of samples, mean intensity 53AU. The molecular mass of the total IgG3 Hc in the plasma samples of the COVID-19 ARDS patients’ convalescent plasma was significantly higher than the other groups (Mean difference: Δ200 *m*/*z*, +0.37% change in mass, *p* = 0.00236) (see Figure 2).

### 3.3. Direct Plasma HSA Spectra

Human albumin was detected in all samples and no statistically significant variance in albumin intensity could be found in these post infection convalescent plasma samples. However, the molecular mass of the convalescent plasma HSA in the patients, who were recovering from ARDS as a result of SARS-CoV-2, was statistically significantly larger than total HSA found in HCWs who had recovered from SARS-CoV-2 infection with only mild symptoms: Mean difference Δ50 *m*/*z*, +0.08% change in mass, *p* = 0.00758).

## 4. Discussion

The total HSA in patients who had developed ARDS COVID-19 had an average molecular weight higher than those developing only mild symptoms, which is a significant potential marker finding. This was not entirely surprising, given that the elderly and particularly those with diabetes are most at risk [20]; and our previous study that the prefusion complete spike protein binds AGE/glycated albumin [14]. Thus, it is entirely consistent with the finding that higher molecular weight glycated albumins are at such a high proportion in unextracted plasma in those who had developed ARDS as a result of COVID-19.

More surprising was that, when detected, the MALDI-ToF mass spectra of total IgG3 Hc was also of a significantly higher mass in the COVID-19 ARDS convalescent plasma (see Figure 2). It had been reported that a dominance of anti-SARS-CoV-2 IgG3 occurs in the immune responses to SARS-CoV-2 in those who died from COVID-19 [16]. We had shown an increase in the mass of IgG3 Hc bound to SARS-CoV-2 prefusion spike protein and eluted from magnetic bead to which it was conjugated [15]. However, to find a dominance of this higher molecular weight IgG3 Hc, in total plasma from patients who had recovered from COVID-19 ARDS, indicates this is probably a characteristic of their inherent immunoglobulin synthesis, i.e., hyper glycosylation [21,22]. Further to being a potential measurable risk marker, the magnitude of the mass difference at ~200 *m*/*z* is larger than that previously encountered when extracting SARS-CoV-2 antigen-bound antibodies (medium increase ~150 *m*/*z*), and better resolved [15].

This increased resolution of a higher mass/hyper-glycosylated total IgG3 Hc, compared to when specific IgG3 capture and elution was analysed by MALDI-ToF, could be explained by terminal glycan sugar hydrolysis and loss from the N- and O- linked moieties. Sialic acid is a common terminal structure on all glycans; however, several factors result in loss of detectable sialic acid residues including low pH and/or high temperature [23,24,25,26,27]. This is particularly common in MALDI-ToF analysis, with acidic matrices (such as Sinapinic Acid (SA)) [28,29]; and certainly, will have occurred during extraction of anti-SARS-CoV-2 immunoglobulins from antigen-bound magnetic beads with pH2 5% acetic acid [14,15]. Indeed, we have noted that when recovering bound albumin, 5% acetic acid was sufficient to hydrolyse the N-terminal aspartic acid residue from the albumin, resulting in a mass decrease of ~115 *m*/*z* compared to HSA not exposed to 5% acetic acid. Such prolonged acetic acid hydrolysis at elevated temperatures will selectively cleave at all aspartic acid residues within a protein [30].

Terminal sialylation of glycan branches have been implicated in many regulatory processes of glycoproteins, including circulatory half-life [31]. Terminal sialylation of O-linked glycans of the carboxy terminal region of the beta-subunit of human chorionic gonadotropin (hCG) are responsible for this hormone’s increased blood circulatory half-life to 24 h, compared to its 82% homologous hormone Luteinising hormone, which has a half-life of only ~20 min [32,33,34]. IgG3 has a variable reported half-life and is the only IgG subtype to have serine O-linked glycosylation sites within the defining neck region of the common structure [35] (see Figure 3). The variability and extent to which these serine residues of the IgG3 neck domain are glycosylated and terminally sialylated, may be decided by other factors including a genetic preference. Similarly, IgG3 has an additional potential N-linked glycosylation site within the neck domain which may or may not be processed [35]. Furthermore, tri-antennary rather than bi-antennary branched glycans are a common feature of variant glycosylation [36,37]. Thus, the loss of terminal sialic acids and other chain saccharides by acidic hydrolysis may mean that only residual branching sugars in extracted and purified immunoglobulin peptide chains are left to reflect the true extent of glycan variances [26]. The direct analysis of unextracted plasma by MALDI-ToF circumvents sialic acid and other glycan losses, due to acid hydrolysis, which are an unintended consequence of purification. Thus, this explains the much clearer size differences in IgG3 Hc revealed in direct plasma or serum sample analysis by MALDI-ToF mass spectrometry.

The importance of sialic acid residues as a target for attachment of Influenza is well known [38,39]. However, human coronaviruses OC43 and HKU1 also bind to 9-*O*-acetylated sialic acids via a conserved receptor-binding site in spike protein domain A [40]. Found within the N-terminal Domain (NTD) of S1 subunit of SARS-CoV-2, this sialic acid residue binding domain has been postulated to be the key feature of coronavirus zoonotic transfer [41]. Its role in corona virus attachment as an airborne to respiratory mucosal epithelia is now recognised [42,43,44,45]. Furthermore, so strong and specific is the Spike complexes binding to sialic acid that one group has used terminally sialylated glycans as a solid phase for SARS-CoV-2 capture in a lateral flow diagnostic device [46]. It is not a large binding domain and may be a charge specific attraction. Terminal sialic acids confer a strong net negative charge, whilst other terminal glycan saccharides do not. Similarly, some of the AGE glycation products deposit a structured negative charge to albumin [47]. This may be a major component of the biochemistry of glycated HSA and hyperglycosylated (and/or glycated) IgG3 binding by SARS-CoV-2 prefusion Spike protein (see Figure 4).

The human cost of the COVID-19 pandemic has been nearly four million lives lost worldwide. The financial cost to the USA is estimated at more than $16 trillion, or approximately 90% of the annual gross domestic product (GDP) of the US [48]. In the UK, borrowing (deficit) in 2021 grew by £300 billion, directly attributable to COVID-19 and in 2020 GDP decreased by 11.3% [49]. With estimates of gross domestic product lagging behind pre-pandemic forecast by 3% annually for many years, the final cost for the UK maybe a similar 90% of GDP. If this is reflected globally then 90% of the global GDP will put the final cost at $76 trillion.

Clearly the investment in medical technologies to detect the virus, prevent its infection and identify those at most risk from COVID-19, along with future viral outbreaks, is both an ethically and financially imperative. In this respect clinical mass spectrometry has only just started to indicate its utility [50,51].

The potential molecular mechanisms, by which sialic acid glycosylation and alike glycations contribute to the pathogenesis of SARS-CoV2, require further investigation (see Figure 4).

Nevertheless, the direct detection of AGE glycated HSA and high mass (hyperglycosylated and or glycated) IgG3 Hc as a rapid biomarker screening test, for the identification of those most at risk of developing life-threatening ARDS as a result of SARS-CoV-2 infection, is a compelling possibility.

## Figures and Tables

**Figure 1 diagnostics-12-02521-f001:**
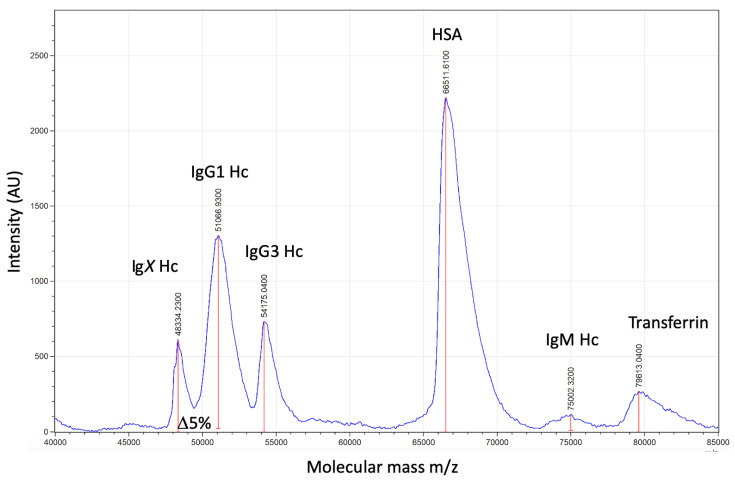
Mass spectra profile, 40,000 *m*/*z* to 85,000 *m*/*z*, of human reduced plasma samples (treated with Tris(2-carboxyethyl)phosphine (TCEP)) in order to reveal immunoglobulin heavy chains (IgM Hc ‘75,000 *m*/*z*, IgA ~56,000 *m*/*z* not found, IgG3 54,000 *m*/*z* and unconfirmed IgX, thought to be IgG4 Hc, at 48,000 *m*/*z*. HSA resolved at 66,4000+ *m*/*z* and transferrin at 79,000 *m*/*z*.

**Figure 2 diagnostics-12-02521-f002:**
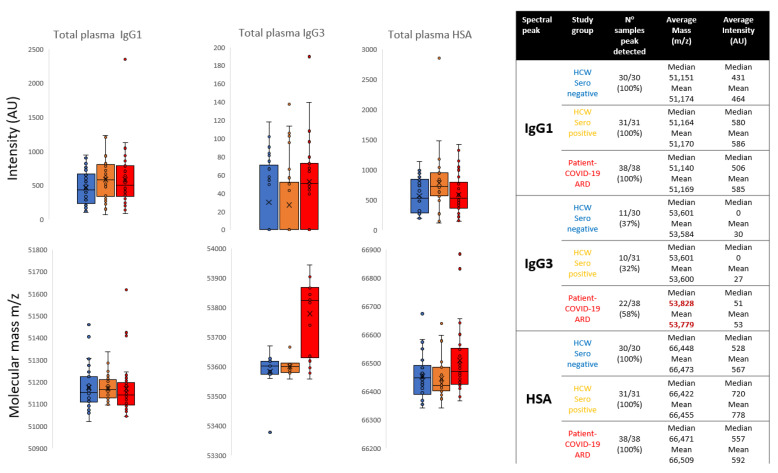
Box and whisker plots of relative intensitities and variance in peak apex molecular mass of IgG1 heavy chains (IgG1 Hc), IgG3 heavy chains (IgG3 Hc) and human serum albumin (HSA) for the different sample groups: Blue represents data from SARS-CoV-2 seronegative HCWs, Orange from SARS-CoV-2 seropositive HCWs having recovered from COVID-19 with mild symptoms and Red sample data from convalescent patients recovering from COVID-19 ARDS. The table to the right is numeric data from the plots detailing the mean and median values of peak intensity (arbitrary units (AU)) and molecular mass (*m*/*z*) of IgG1 Hc, IgG3 Hc and HSA for the respective groups.

**Figure 3 diagnostics-12-02521-f003:**
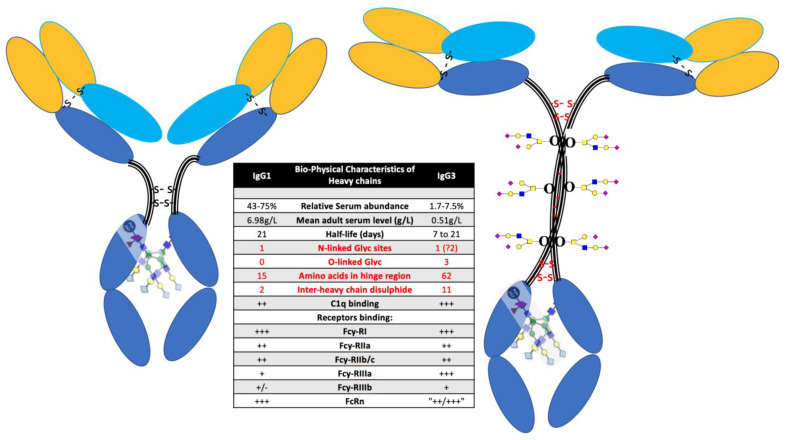
Diagrammatic representation of IgG1 (**left**) and IgG3 (**right**) illustrating the differences in heavy chain structure with special reference to the larger neck domain and the O-linked glycosylation sites found there. Affinity binding: –, none; +,weak; ++, strong; and +++, very strong.

**Figure 4 diagnostics-12-02521-f004:**
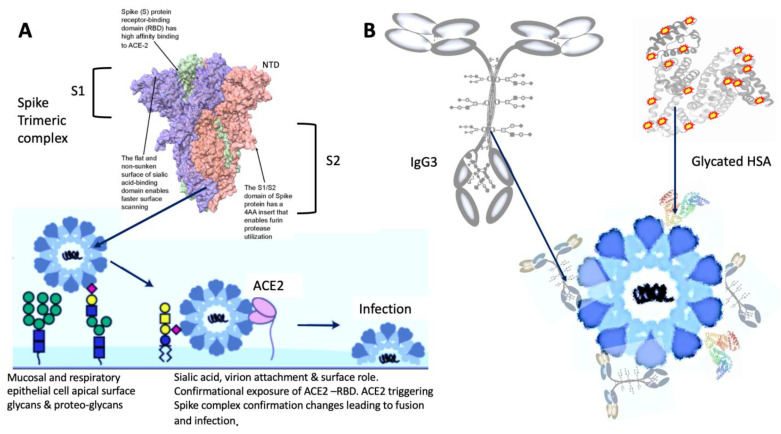
Spike protein sialic acid glycan binding (S1 subunit NTD-domain) role in (**A**) mucosal epithelial attachment and infection of respiratory cells and (**B**) potential role in binding IgG3 at the neck region and AGE glycated HSA, during viremia, thereby aiding immune evasion and deposition of reactive complex that may give rise to vascular pathologies.

## Data Availability

Compiled summary data can be made available upon request to the corresponding author. Raw mass spectral data from the individual samples will require compiling from archives at MAP Sciences and so require a detailed project proposal to justify the additional resource expenditure required in providing this complete data set.

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
