# Peer review of "Direct Detection of Glycated Human Serum Albumin and Hyperglycosylated IgG3 in Serum, by MALDI-ToF Mass Spectrometry, as a Predictor of COVID-19 Severity"

_diagnostics, 2022, doi:10.3390/diagnostics12102521_

Round 1

Reviewer 1 Report (Previous Reviewer 2)

The authors have made necessary corrections and the manuscript can be published in its current form. 

Author Response

We thank the author very much for their approval

Reviewer 2 Report (New Reviewer)

The authors Iles et al, have presented a small study on SAR-CoV-2 comparing 30 seronegative and 31 seropositive health workers against 38 patients with ARDS associated with COVID-19. The results show significantly higher mass of AGE-glycated HSE and IgG3 in patients recovering from COVID infection. The implications of these results if validated may be useful in how we tackle future outbreaks and screen effectively.

Some comments for the authors:

1. How does the method normalise to biomass? It is unclear in the spectral data processing. If it is assumed same biomass based on volume of sample, this should be stated to negate any need of statistical normalisation.

2. Are Mann Whitney U test, probability values corrected for false discovery rates or multiple testing? It is unclear in the current write up. Please clarify.

3. Box plot in Fig 2 shows that comparisons are likely between raw intensities of these features. Could these be compared as log normalised intensities to scale them appropriately?

4. Can the peak(s) with statistical differences be modelled using ROC and be represented in terms of specificity and sensitivity? 

5. Fig 3 has a table that is very tiny and is hard to be read. Can this be enlarged or an enlarged version added to supplementary information? The table is of importance to this work but in the current format it is hard to read.

6. Fig 4a looks stretched sideways and not publication quality. Can this be re-done or re-sized to remove the stretch?

7. List of abbreviation: MALDI-ToF MS, doesn't have full form for MS. Either remove MS from abbreviation or define this in the list please.

8. The word 'biomarker' is used often in the manuscript but not much beyond a t-test has been performed. I suggest calling this a potential biomarker or potential marker for COVID-19 as much needs to be added to the study for it to truly be a biomarker. 

9. Why is HCW detected in only 32-37% of seronegative and seropositive workers? This is unclear from the results and discussion section. Is this something observed in other similar studies? Adding relevant biological context will improve the discussion. 

10. Fig1, axes labels are too tiny, please enlarge them to make it visible. 

Author Response

We very much thank the reviewer for their constructive comments. Our responses are below.

1. How does the method normalise to biomass? It is unclear in the spectral data processing. If it is assumed same biomass based on volume of sample, this should be stated to negate any need of statistical normalisation.

This is indeed a question we considered. We had estimated the total protein of the serum samples but in routine clinical chemistry serum sample analytes are not corrected to total sample protein, just to the volume of sample. Here it was 1ul of serum/plasm sample diluted  in 79ul of ddH20 AND then 1ul of that dilutuant plated on to the mass spectrometer matrix plate. Thus, biomass is based on the volume of serum/plasma but indirectly via a dilution step as detailed in the material and methods.

2. Are Mann Whitney U test, probability values corrected for false discovery rates or multiple testing? It is unclear in the current write up. Please clarify. 

Yes this is was correctly applied. The Mann Whitney U test were applied to specific comparisons showing a graphic distribution difference, not random and not to all measures regardless. 

3. Box plot in Fig 2 shows that comparisons are likely between raw intensities of these features. Could these be compared as log normalised intensities to scale them appropriately?

We have applied both log and linear scale analysis however linear scaling was the most informative in this particular direct serum/plasma sample analysis. 

4. Can the peak(s) with statistical differences be modelled using ROC and be represented in terms of specificity and sensitivity? 

This will be the subject of a subsequent study of a much larger cohort. Here the test is not of who would go onto develop COVID-19 but who had developed COVID-19. Thus, the biggest problem we have in respect to generating a meaningful ROC prediction is we would have to have had blood samples from patients presenting before they developed COVID-19 and these are samples after recovery. Although as clearly demonstrated it shows the fundamental difference in those infected with and recovered from SARS-CoV-2, who did and did not develop ARDS. 

5. Fig 3 has a table that is very tiny and is hard to be read. Can this be enlarged or an enlarged version added to supplementary information? The table is of importance to this work but in the current format it is hard to read.

Yes this is being adapted

6. Fig 4a looks stretched sideways and not publication quality. Can this be re-done or re-sized to remove the stretch?

YES

7. List of abbreviation: MALDI-ToF MS, doesn't have full form for MS. Either remove MS from abbreviation or define this in the list please.

MS has been removed

8. The word 'biomarker' is used often in the manuscript but not much beyond a t-test has been performed. I suggest calling this a potential biomarker or potential marker for COVID-19 as much needs to be added to the study for it to truly be a biomarker. 

Agreed biomarker has been changed to potential marker

9. Why is HCW detected in only 32-37% of seronegative and seropositive workers? This is unclear from the results and discussion section. Is this something observed in other similar studies? Adding relevant biological context will improve the discussion. 

This has been changed to give clarity

10. Fig1, axes labels are too tiny, please enlarge them to make it visible. 

This has been adjusted.

This manuscript is a resubmission of an earlier submission. The following is a list of the peer review reports and author responses from that submission.

Round 1

Reviewer 1 Report

Comments

Ray K Iles et al., describe the identification of humoral markers for Covid 19 using MALDI-ToF mass  spectrometry and have described gly- 2 cated HSA and hyperglycosylated IgG3 as humoral markers.  I think this study is much needed and therefore, undoubtedly it is the need of the hour. However, there are specific things that need to be addressed and elaborated before I would recommend this for publication subject to the following major revisions.

In the introduction section, the authors should briefly explain the role and significance of each immunoglobulin in general and with regard to COVID 19 that would provide the readers an easier understanding of the technique used.

The authors are also asked to emphasize the advantages of Mass spectrometry, in general and MALDI ToF MS, in particular, in disease diagnosis and detection.

In Fig.1 Abbreviate the Y axis label. Although the peaks in the mass spectrum appears to be good, while the intensity of the ion peak in the Y axis are very low, >2500 ions. Therefore, the authors should improve the quality of the spectrum by optimizing the sample (protein) quantity.

In line number 137, authors wrote “IgG3 Hc mass spectral peaks were detected in 43/99…”.  I recommend the authors to mention the peak value (m/z value) of IgG3 HC in parenthesis and follow that for all the immunoglobulins throughout the manuscript with their respective values (m/z values).

Provide the representative Mass spectra describing the peak shift for the biomarker immunoglobulins and HSA as supporting figure

Reviewer 2 Report

Though COVID 19 incidence is declining there is still a need for fast and relevant methods of its diagnosis and prediction of possible complications. On the one hand the proposed approach looks rather attractive because of simple and fast sample preparation procedures and analysis. But on the other one the manuscript doesn't look containing enough novelty to be published. The glycosylation of immunoglobulins as a result of COVID 19 has already been described several times [see, for example, 10.1016/j.ebiom.2022.103957]. At the same time the authors just stated the differences in IgG profiles in MALDI mass spectra and haven't performed any tryptic experiments to determine types and site of glycosylation. In my opinion it is not enough for publishing.  So I recommend making additional experiments and resubmitting the manuscript after addition of this data.